# Cardiometabolic diseases and associated risk factors in transitional rural communities in tropical coastal Ecuador

Monsermin Gualan[1,2☯], Irina Chis Ster[3☯], Tatiana Veloz[1], Emily Granadillo[1,2], Luz M. Llangari-Arizo[1,2], Alejandro Rodriguez[1], Julia A. Critchley[4], Peter Whincup[4], Miguel Martin[2,5], Natalia Romero-Sandoval[1,2], Philip J. Cooper[1,3]*

1 School of Medicine, Universidad Internacional del Ecuador, Quito, Ecuador, 2 Grup's de Recerca d' Amèrica i Àfrica Llatines-GRAAL, Barcelona, Spain, 3 Institute of Infection and Immunity, St George's University of London, London, United Kingdom, 4 Population Health Research Institute, St George's University of London, London, United Kingdom, 5 Unidad de Bioestadística, Facultad de Medicina, Universidad Autónoma de Barcelona, Barcelona, Spain

☯ These authors contributed equally to this work.
* pcooper@sgul.ac.uk

## Abstract

### Background

There is a growing epidemic of chronic non-communicable diseases in low and middle-income countries, often attributed to urbanization, although there are limited data from marginalized rural populations. This study aimed to estimate prevalence of cardiometabolic diseases and associated risk factors in transitional rural communities.

### Methods

A cross-sectional study of Montubio adults aged 18–94 years living in agricultural communities in a tropical coastal region of Ecuador. Data were collected by questionnaires and anthropometry, and fasting blood was analyzed for glucose, glycosylated hemoglobin, insulin, and lipid profiles. Population-weighted prevalences of diabetes, hypertension, and metabolic syndrome were estimated. Associations between potential risk factors and outcomes were estimated using multilevel regression techniques adjusted for age and sex.

### Results

Out of 1,010 adults recruited, 931 were included in the analysis. Weighted prevalences were estimated for diabetes (20.4%, 95% CI 18.3–22.5%), hypertension (35.6%, 95% CI 29.0–42.1%), and metabolic syndrome (54.2%. 95% CI 47.0–61.5%) with higher prevalence observed in women. Hypertension prevalence increased with age while diabetes and metabolic syndrome peaked in the 6th and 7th decades of life, declining thereafter. Adiposity indicators were associated with diabetes, hypertension, and metabolic syndrome.

### Conclusion

We observed an unexpectedly high prevalence of diabetes, hypertension, and metabolic syndrome in these marginalized agricultural communities. Transitional rural communities

**Data Availability Statement:** All relevant data are within the manuscript and its Supporting Information files.

**Funding:** Universidad Internacional del Ecuador (grant EDM-INV-04-19). The study funders had no role in study design, data collection and analysis, decision to publish, or preparation of the manuscript.

**Competing interests:** The authors have declared that no competing interests exist.

are increasingly vulnerable to the development of cardiometabolic risk factors and diseases. There is a need for targeted primary health strategies to reduce the burden of premature disability and death in these communities.

## Introduction

Low and middle-income countries (LMICs) are undergoing the epidemiologic transition with a shift in the primary causes of death from those attributable to infectious and childhood diseases to chronic non-communicable diseases (NCDs) in the context of rapidly aging populations [1, 2]. Among NCDs, age-adjusted prevalence of cardiovascular diseases peaked over the period 1960 to 1980 in high-income countries (HICs), while an epidemic of these diseases has now emerged in LMICs [3, 4], which now account for more than 77% of all deaths attributed to NCDs globally [5].

The large increases in prevalence of NCDs in LMICs have been attributed to economic development and urbanization that have resulted in dramatic changes in where and how people live [6] with consequent changes in living conditions, diet, physical activity, and exposures to psychosocial stressors [7]. The emergence of the NCD epidemic in LMICs is having a major impact on utilization of scarce health care resources and poses a significant hurdle to achieving sustainable development goal targets for poverty reduction and prevention of premature death and disability [5].

Ecuador is an upper middle-income country in Latin America [8] with a diverse multiethnic population of 17 millions [9], life expectancy of 74 years, and a per capita gross domestic product of US$6,395 in 2022 [10]. NCDs are now the major causes of death in Ecuador: among the five most important causes of death reported in 2019, 4 were NCDs including ischemic heart disease (14.7% of deaths), type 2 diabetes (T2D) (7.1%), cerebrovascular diseases (6.2%), and hypertension (HTN) (4.9%) [11].

The NCD epidemic is increasingly present in rural populations [12] through the influence of urbanization that extends even to the most isolated communities [13–15]. While a rural lifestyle is presumed to reduce the risk of many NCDs because of potentially protective factors—such as consumption of traditional diets high in complex carbohydrates, increased physical activity, and greater social integration [16]—in many rural areas such protective exposures are being lost. Indigenous and other vulnerable population groups living in transitional settings undergoing environmental degradation, urbanization, and the loss of traditional lifestyles, are at increased risk of cardiometabolic diseases [17].

We carried out a cross-sectional study of adults living in transitional rural communities of Montubios, a marginalized mestizo-derived ethnicity, in an ecologically vulnerable region of coastal Ecuador to estimate prevalence of cardiometabolic NCDs and associated risk factors. Our findings, showing a high prevalence of cardiometabolic diseases, highlight an unmet need for community-based public health strategies to minimize the growing burden of premature death and morbidity from NCDs among marginalized and indigenous populations living in transitional rural settings.

## Methods

### Study design and population

We performed a cross-sectional study of adults living in ten adjacent rural agricultural communities in Abdon Calderon parish, Portoviejo district, Manabí province, in tropical coastal

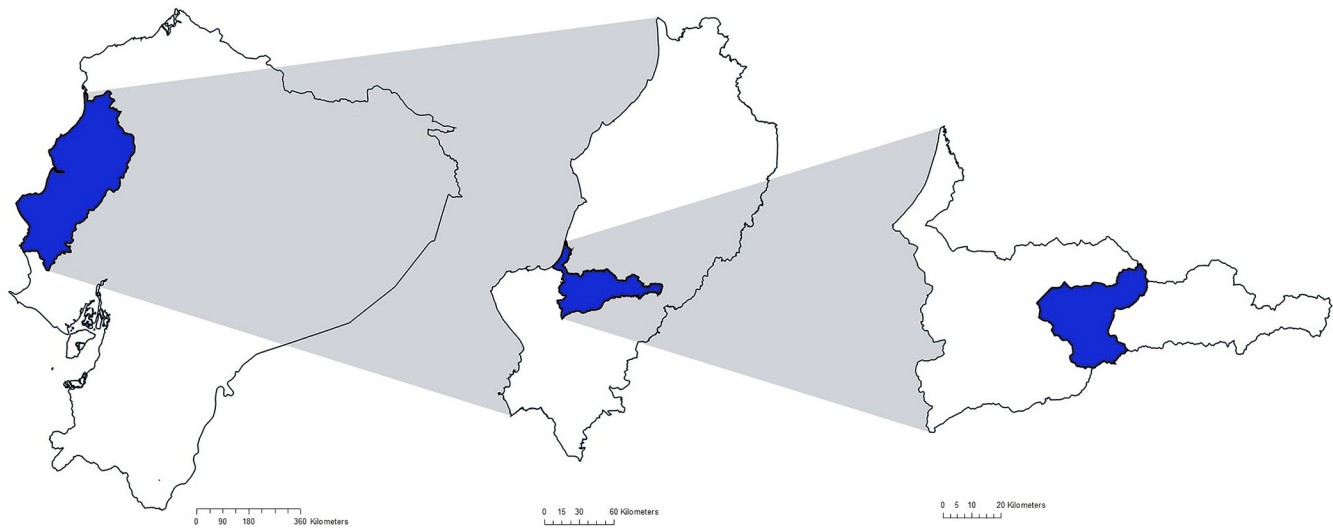

**Fig 1. Geographical context of the study area showing from left to right, locations in Ecuador of Manabí Province, Portoviejo District, and Abdon Calderon Parish.** The figure was designed using shapefiles obtained from the Ecuadorian National Institute of Statistics (INEC), (https://www.ecuadorencifras.gob.ec/documentos/web-inec/Geografia_Estadistica/Micrositio_geoportal/index.html#cartograf-histor) and constructed using QGIS version 3.14.

Ecuador. The study area (Fig 1) was traditionally inhabited by Montubios, a Spanish-speaking mestizo-derived ethnicity, geographically restricted to the mountainous regions of coastal Ecuador, and who possess distinct socio-cultural characteristics that distinguish them from the dominant mestizo population. Montubios, along with Indigenous and Afro-Ecuadorians, belong to the poorest ethnic groups in Ecuador and live in communities with important unmet health needs [18]. The study communities were located 10 km from a primary health care facility and 30 km from a regional hospital, in a region of Abdon Calderon parish considered to be highly vulnerable to environmental degradation, and hence deprived of access to services and infrastructure investment. Initial contacts were made with the parish council through local political leaders known to the study team and who acted as gatekeepers. Community representatives were involved at all stages of the study including extensive early meetings to inform study design, objectives, and selection of study site; organisation of meetings to explain study objectives and obtain community consent for the study; planning and organization for individual consent and data collection as well as involvement in the data collection process itself; and community meetings to provide individual results and explain study findings. Communities were selected by convenience sampling. Censuses for each community were drawn up by community representatives specifically for this study. All adults aged 18 years and older living in the communities were invited to participate through community assemblies. Exclusions were those unwilling to provide informed written consent, pregnant women or those having given birth in the previous 3 months. Recruitment took place during scheduled visits to study communities between 6 and 10 am from 31st July to 12th September 2021.

### Data collection

Data and clinical samples were collected, and clinical measurements performed as follows using standardized protocols: i) questionnaire based on STEPwise approach to NCD risk factor surveillance (STEPS) [19] to collect data on symptoms, treatments and diagnoses, and potential risk factors; ii) anthropometric measurements including weight (TANITA model BF-

60W, Japan), height (SECA model 213, Germany), waist and hip circumferences, and bioimpedance (Bodystat model 1500, UK); iii) blood pressure seated after 15 minutes rest using electronic devices (OMRON model 7051T, Japan) with daily calibration against a mercury sphygmomanometer and cuffs selected according to upper arm circumference. This was repeated after 5 minutes if systolic blood pressure (SBP) >130 mmHg and/or diastolic blood pressure (DBP) >85 mmHg; and iv) blood after an overnight fast to measure glucose, insulin, glycosylated hemoglobin (HbA1c), cholesterol, high-density lipoprotein (HDL), and triglycerides.

## Laboratory analyses

HbA1c was measured with fresh anticoagulated venous blood using a fluorescent immunoassay at a field laboratory (iChroma™, Boditech Med Incorporated, South Korea). Serum and plasma were separated by centrifugation and stored at -20C for later analysis at a central laboratory in Quito. Biochemical analyses (cholesterol, HDL, and triglycerides) were done using enzymatic colorimetry assays (Human Diagnostics, Germany). Insulin was analyzed using an immunoenzymatic assay (DIAsource ImmunoAssays, Belgium).

## Definitions

Definitions for outcomes were: i) Type 2 diabetes (T2D)—clinical history of diabetes treatment and/or glycosylated hemoglobin (HbA1c) ≥6.5% [20]; ii) hypertension (HTN)–clinical history of antihypertensive treatment and/or SBP ≥140 mm Hg or diastolic blood pressure DBP ≥90 mm Hg during 2 separate measurements [21]; and iii) metabolic syndrome (MetS)– 3 or more of the following (using the Harmonized criteria [22]: a) elevated waist circumference of ≥94 cm in men and ≥88 cm in women [23]; b) triglycerides ≥150 mg/dL or treatment with triglyceride-lowering drugs; c) reduced high density lipoprotein (HDL) <40 mg/dL in men and <50 mg/dL in women; d) elevated blood pressure ≥130/85 mmHg or treatment with antihypertensives; and e) fasting glucose ≥100 mg/dL or treatment with glucose-lowering drugs.

Definitions for other conditions were: i) Adiposity indicators [24–26]: a) overweight (body mass index ≥25) and obesity (BMI ≥30); b) abdominal obesity—waist circumference of ≥94 cm in men and ≥88 cm in women [23]; c) elevated weight-to-height (WHtR)—≥0.5; d) body fat (free fat mass) was determined using bioimpedance [27] with increased body fat defined as ≥25% for men and ≥30% for women; e) visceral adiposity index (VAI) was calculated using a model of adipose distribution corrected for triglyceride and HDL levels [23] with high VAI according to age defined as—age <30 –VAI >2.52; age ≥30 & <42 –VAI >2.23; age ≥42 & <52 –VAI >1.92; age ≥52 & <66 –VAI >1.93; age ≥66 –VAI >2; ii) elevated cholesterol ≥200 mg/dl; iii) elevated triglycerides ≥150 mg/dL; iv) low HDL <40 mg/dl in men and <50 mg/dl in women; v) insulin resistance–Homeostatic Model Assessment (HOMA) index >2.5 [28, 29]; and vi) and previous medical diagnosis (personal history) of heart attack or stroke. Occupation was categorised into 3 categories: agricultural workers, non-agricultural workers, and household chores. Ethnicity was self-identified and categorised into two groups: mestizos and non-mestizos, the latter including Indigenous, Afro-Ecuadorians, Montubios, and White.

## Statistical analysis

T2D, HTN, and MetS were analysed as binary outcomes, while blood pressure, glycemia, and HbA1c were analysed as continuous variables. Because of the three-level hierarchical structure of the data (individuals are nested in households and communities), multilevel regression techniques were used to estimate standard errors and P values. We used age and sex data from the census population in district of Portoviejo (S1 Fig) [9] to derive post-stratification weights to inflate misrepresented groups and reduce effect of overrepresented groups in the sample [30].

We derived prevalences (T2D, HTN, and MetS) and means (SBP, DBP, HbA1c and glycemia) stratified by sex, age (up to power of three terms, i.e. non-linear), and their interactions when statistically significant (P<0.05). We estimated age- and sex-adjusted effects of explanatory variables of interest on prevalence of T2D, HTN, and MetS. Sensitivity analyses were performed for missing data but showed consistent findings to those obtained using complete data. Analyses were exploratory and adjusted only for age and sex rather than the construction of multivariable models. All analyses were done using Stata 18 (version 18, Statacorp, College Station, Texas, USA).

### Ethics statement

Informed written consent was obtained from each participant and the study protocol was approved by Bioethics Committee of the UTE University (Comité de Ética de Investigación en Seres Humanos de la Universidad UTE, CEISH UTE 2019-1121-03) and by the Ecuadorian Ministry of Public Health (MSP-DIS-2021-0393-O).

## Results

### Study population and characteristics

Of 1,525 adults eligible according to the house-to-house census, 1,010 adults were recruited from 10 communities of which 931 (61.0% of those eligible) provided complete data and were included in the analysis. Total and sex-stratified data on socio-demographic factors, indicators of adiposity, biochemical measurements, and clinical history of NCDs are shown in Table 1. The mean age of the study population was 44.7 years (standard deviation 18.1) and 57.5% were women. Most participants lived with a partner (66.1%); 24.3% were functionally illiterate; 48.3% self-identified as non-mestizo with 41.1% identifying as Montubio, 2.3% as Afro-Ecuadorian, and 0.5% as Indigenous. Most (62.6%) men were agricultural workers while 80.9% of women were housewives; recent alcohol consumption was reported by 62.2% (men 77.6% vs. women 50.8%); current smoking was reported by 7.7% (men 17.2% vs. women 0.8%); men were more likely to do physically demanding work and have contact with agrochemicals. Indicators of adiposity were more frequent in women (for example, presence of abdominal obesity [men 49.5% vs. women 66.4%] and elevated body fat [men 93.1% vs. women 97.9%]). Insulin resistance affected 58.6% of the study population and was more frequent in women. Clinical histories of vascular events were reported by 8.7%. Of those with a clinical history of diabetes or hypertension, only 25.8% had evidence of adequate glucose control (S1 Table) and 34.9% had controlled blood pressure (S2 Table), respectively. A comparison of participation and non-participation using our house-to-house census data showed an increase in participation with age (OR 1.02, 95% CI 1.01–1.03) and among women (OR 5.4, 95% CI 3.7–7.8).

### Population-weighted prevalence of diabetes, hypertension, metabolic syndrome, and vascular events

Table 2 shows the crude and population-weighted prevalence of outcomes. Weighted prevalence for T2D, HTN, and MetS were higher among females. Age-prevalence profiles for the 3 outcomes by sex are shown in Fig 2 and S1 Table. For T2D, prevalence was greater in women at all ages (Fig 2C) and increased with age reaching a peak of 51% in women and 37% in men at around 75 years, after which it declined. Prevalence of HTN increased with age although there was some evidence of an interaction with sex, particularly showing a steep increase in prevalence in women (Fig 2B). For MetS, prevalence increased in both sexes until 57 years in men (70%) and 69 years in women (86%) after which prevalence declined (Fig 2A). Prevalence

**Table 1. Baseline frequencies of age and risk factors for chronic non-communicable diseases stratified by sex in 931 adults in a rural parish of Manabi Province, Ecuador.**

| Variable | Category | Total (n = 931) n (%) | Women (n = 535) n (%) | Men (n = 396) n (%) |
|---|---|---|---|---|
| **Age distribution and risk factors** | | | | |
| Age group (years) | 18–29 | 229 (24.6) | 131 (24.5) | 98 (24.8) |
| | 30–39 | 158 (17.0) | 100 (18.7) | 58 (14.6) |
| | 40–49 | 183 (19.7) | 107 (20.0) | 76 (19.2) |
| | 50–59 | 149 (16.0) | 82 (15.3) | 67 (16.9) |
| | 60–69 | 104 (11.2) | 56 (10.5) | 48 (12.1) |
| | ≥ 70 | 108 (11.5) | 59 (11.0) | 49 (12.4) |
| Lives with partner | Yes | 615 (66.1) | 362 (67.7) | 253 (63.9) |
| Functional illiteracy | Yes | 226 (24.3) | 114 (21.3) | 112 (28.3) |
| Ethnicity | Non-mestizo | 450 (48.3) | 229 (42.9) | 221 (55.8) |
| | Mestizo | 481 (51.7) | 306 (57.2) | 175 (44.2) |
| Occupation | Agricultural worker | 261 (28.1) | 13 (2.4) | 248 (62.6) |
| | Household chores | 436 (46.8) | 433 (80.9) | 3 (0.8) |
| | Non-agricultural workers | 234 (25.1) | 89 (16.6) | 145 (36.6) |
| Recent alcohol consumption | Yes | 575 (62.2) | 270 (50.8) | 305 (77.6) |
| Current smoker | Yes | 72 (7.7) | 4 (0.8) | 68 (17.2) |
| Physically-demanding work | Yes | 319 (34.6) | 104 (19.6) | 215 (54.8) |
| Contact with agrochemicals | Yes | 253 (27.2) | 47 (8.8) | 206 (52.0) |
| **Indicators of adiposity and biochemical risk factors** | | | | |
| Obesity | Not overweight (BMI <25) | 322 (34.6) | 164 (30.5) | 159 (40.2) |
| | Overweight (BMI 25–29) | 367 (39.4) | 212 (39.6) | 155 (39.1) |
| | Obese (BMI ≥30) | 242 (26.0) | 160 (29.9) | 82 (20.7) |
| Abdominal obesity | men ≥94cm; women ≥88cm | 551 (59.2) | 355 (66.4) | 196 (49.5) |
| Elevated WHtR | ≥0.5 | 791 (85.0) | 475 (88.8) | 316 (79.8) |
| Elevated body fat | men (≥25%); women (≥30%) | 696 (95.9) | 412 (97.9) | 284 (93.1) |
| Elevated VAI | See legend | 571 (61.3) | 368 (68.8) | 203 (51.3) |
| Elevated cholesterol | ≥200 mg/dl | 445 (47.8) | 270 (50.5) | 175(44.2) |
| Elevated triglycerides | ≥150 mg/dL | 434 (44.6) | 257 (48.0) | 177 (44.7) |
| Low HDL | Men <40 mg/dl; women<50 mg/dl | 524 (56.3) | 367 (68.6) | 157 (39.7) |
| Insulin resistance | HOMA>2.5 | 546 (58.6) | 372 (69.5) | 174 (43.9) |
| Elevated HbA1c | ≥6.5% | 210 (22.6) | 133 (24.9) | 77 (19.4) |
| **Self-report of non-communicable diseases or risk factors (history of disease)** | | | | |
| Vascular events (stroke or heart attack) | Yes | 81 (8.7) | 42 (7.9) | 39 (9.9) |
| Diabetes | Yes | 133 (14.3) | 95 (17.8) | 38 (9.7) |
| Hypertension | Yes | 305 (32.9) | 200 (37.4) | 105 (26.7) |
| Elevated cholesterol | Yes | 364 (39.2) | 237 (44.3) | 127 (32.3) |

Functional illiteracy was defined as 3 or fewer years of formal schooling [29]. Abbreviations: WHtR–waist-to-height ratio; VAI–visceral adiposity index; HDL–high-density lipoprotein. High VAI—age <30 –VAI >2.52; age ≥30 & <42 –VAI >2.23; age ≥42 & <52 –VAI >1.92; age ≥52 & <66 –VAI >1.93; age ≥66 –VAI >2. Missing data: alcohol consumption (7); current smoker (1); physical demanding work (8); bioimpedance (increased fat %) (205); history of vascular events (3); history of hypertension (3); history of diabetes (3); and history of elevated cholesterol (3).

of MetS was similar in both sexes up to about 40 years, after which prevalence was greater in women. An analysis of 576 participants, for whom blood pressure could be repeated at home, showed a lower average systolic blood pressure in the home setting (average difference, 7.13 mm Hg [95% CI 5.76–8.51]) but similar diastolic pressure (average difference 0.17 mm Hg [95% CI -0.82–1.16]).

**Table 2. Crude and population-weighted prevalence of chronic diseases and means of blood pressure and glucose measures in study population of 931 adults stratified by sex.**

| Chronic disease | Total (n = 931) | | Women (n = 535) | | Men (n = 396) | |
|---|---|---|---|---|---|---|
| Binary outcomes | Crude % (n) | Estimated weighted % (95% CI) | Crude % (n) | Estimated weighted % (95% CI) | Crude % (n) | Estimated weighted % (95% CI) |
| Type 2 Diabetes | 26.3 (245) | 20.4 (18.3–22.5) | 29.5 (158) | 24.5 (20.9–28.7) | 22.0 (87) | 15.8 (12.2–19.4) |
| Hypertension | 44.7 (416) | 35.6 (29.0–42.1) | 46.2 (247) | 37.7 (31.0–44.4) | 42.7 (169) | 33.3 (26.8–39.7) |
| Metabolic syndrome | 58.0 (540) | 54.2 (47.0–61.5) | 60.9 (326) | 56.2 (47.7–64.7) | 54.0 (214) | 52.1 (43.3–60.8) |
| Vascular events | 8.7 (81) | 4.8 (2.7–6.9) | 7.9 (42) | 4.6 (2.8–6.4) | 9.9 (39) | 5.0 (2.2–7.8) |
| Continuous outcomes | Crude means (n) | Estimated weighted means (95% CI) | Crude means (n) | Estimated weighted means (95% CI) | Crude means (n) | Estimated weighted means (95% CI) |
| Systolic blood pressure (mm Hg) | 134.0 (245) | 131.0 (129.4–132.6) | 132.8 (158) | 129.2 (127.1–131.2) | 135.8 (87) | 132.9 (131.0–134.9) |
| Diastolic blood pressure (mm Hg) | 80.8 (416) | 79.7 (78.8–80.6) | 80.6 (247) | 79.5 (78.3–80.6) | 80.9 (169) | 80.0 (78.7–81.2) |
| Fasting glucose (mg/dL) | 115.0 (540) | 111.1 (108.3–113.8) | 118.8 (326) | 114.6 (110.3–119.0) | 109.9 (214) | 107.3 (104.5–110.1) |
| HbA1c (%) | 6.4 (81) | 6.3 (6.2–6.4) | 6.6 (42) | 6.4 (6.3–6.6) | 6.2 (39) | 6.1 (6.0–6.2) |

Crude sample prevalence and means were derived from clinical history and measurements and weighted estimates (and 95% confidence intervals [CI]) accounted for household and community clustering and were weighted to the census population at district level (S1 Fig). Chronic diseases definitions: i) Type 2 diabetes—clinical history of diabetes treatment and/or glycosylated hemoglobin (HbA1c) ≥6.5%; ii) hypertension–clinical history of antihypertensive treatment and/or SBP ≥140 mm Hg or DBP ≥90 mm Hg during 2 separate measurements [21]; and iii) metabolic syndrome–using the Harmonized criteria [22]; and iv) vascular events- self-reported history of heart attack or stroke.

## Risk exposures associated with diabetes, hypertension, and metabolic syndrome

Age-and sex-adjusted associations between potential risk factors and disease outcome are shown in Table 3. All outcomes were strongly positively associated with age. There were age-sex interactions on the risks of HTN (P = 0.003) and MetS (P = 0.001). Among factors associated with T2D were female sex (male vs. female, adj. OR 0.57, 95% CI 0.41–0.80, P = 0.001), non-agricultural work (vs. agricultural, adj. OR 1.74, 95% CI 1.01–2.98, P = 0.045), HTN (adj. OR 2.09, 95% CI 1.40–3.11), and abdominal obesity (adj. OR 2.85, 95% CI 1.85–4.39). Among factors associated with HTN were household chores (vs. agriculture, adj. OR 3.67, 95% CI 1.71–7.90) or non-agricultural activities (vs. agriculture, adj. OR 1.92, 95% CI 1.13–3.26); history of vascular events (adj. OR 2.45, 95% CI 1.38–4.35); T2D (adj. OR 1.95, 95% CI 1.31–2.90); abdominal obesity (adj. OR 3.12, 95% CI 2.15–4.54); and insulin resistance (adj. OR 3.05, 95% CI 2.10–4.42). MetS was associated with living with a partner (adj. OR 1.55, 95% CI 1.08–2.23), among other factors.

## Associations between indicators of adiposity and measurements of systolic and diastolic blood pressure, fasting glucose, and glycosylated hemoglobin

Means of systolic blood pressure (SBP) and diastolic blood pressure (DBP), fasting capillary blood glucose and glycosylated hemoglobin (HbA1c) in the study population and stratified by sex are shown in Table 2.

There were significant interactions between age and sex for all 4 measurements (P<0.005) (Fig 3 and S3 Table). Predicted values for SBP increased with age being greater in men up to 50 years after which values were greater in women (Fig 3A). Predicted DBP increased with age up to 50 years and then declined, showing a similar sex interaction as for SBP (Fig 3B). Predicted values of fasting glucose increased with age reaching a plateau after 50 years (Fig 3C):

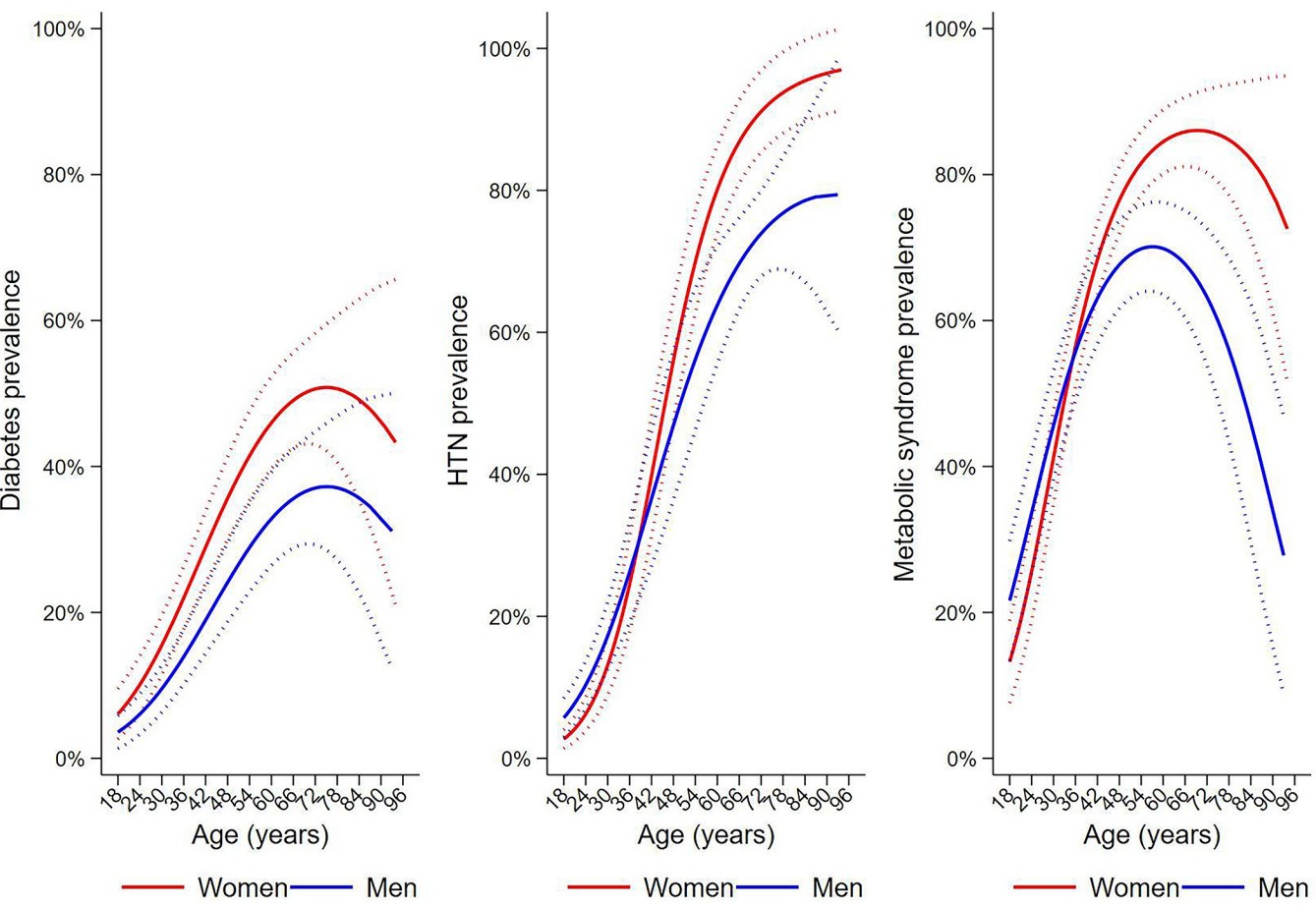

**Fig 2. Metabolic syndrome, HTN and diabetes: The age-dependent prevalence stratified by gender.** Predictions were from logistic regression of the outcomes on age, gender and their potential interactions accounting for family cluster and weighted for population characteristics.

levels were similar in both sexes up to 30 years after which levels diverged becoming greater in women. A similar pattern was seen for HbA1c (Fig 3D). Strong associations were observed between a variety of adiposity indicators and blood pressure levels and glucose measurements (S3 Table).

## Discussion

In this cross-sectional study, we estimated prevalence and associated risk factors for cardiometabolic NCDs in adults living in transitional rural communities in tropical Ecuador. These communities, located in an ecologically vulnerable region of coastal Ecuador, were traditionally inhabited by a marginalized population group, the Montubios. Our data showed a high weighted prevalence of T2D, HTN, and MetS affecting 20.4%, 35.6%, and 54.2%, respectively, of inhabitants in these communities. The majority (74.2%) of those with a doctor diagnosis T2D had poorly controlled disease and 14% were unaware of their condition. Similarly, over one-quarter of those with HTN (26.6%) were unaware of their diagnosis before the study. Over 8% of the study population had a history of vascular events (stroke or heart attack). NCDs were more common in females, and largely increased with age, becoming very common at extremes of age (e.g. 70% with HTN by age 80).

**Table 3. Age-and sex adjusted associations between potential risk factors and type-2 diabetes mellitus, hypertension, and metabolic syndrome in 931 adults.**

| Risk factor | Category | Type-2 Diabetes | | | Hypertension | | | Metabolic Syndrome | | |
|---|---|---|---|---|---|---|---|---|---|---|
| | | n (%) | OR (95% CI) | P value | n (%) | OR (95% CI) | P value | n (%) | OR (95% CI) | P value |
| **Socio-demographic** | | | | | | | | | | |
| Age | | | 1.05 (1.04–1.07) | <0.001 | | 1.10 (1.08–1.11) | <0.001 | | 1.07 (1.06–1.09) | <0.001 |
| Sex | Female | 158 (29.5) | 1 | | 247 (46.2) | 1 | | 326 (60.9) | 1 | |
| | Male | 87 (22.0) | 0.57 (0.41–0.80) | 0.001 | 169 (42.7) | 0.80 (0.58–1.12) | 0.193 | 214 (54.0) | 0.72 (0.53–0.99) | 0.04 |
| Interaction: Age × Sex | | ------ | ------ | ------ | ------ | 0.97 (0.95–0.99) | 0.003 | ------ | 0.97 (0.95–0.99) | 0.001 |
| **Risk exposures** | | | | | | | | | | |
| Lives with partner | No | 67 (21.2) | 1 | | 124 (39.2) | 1 | | 146 (46.2) | 1 | |
| | Yes | 178 (28.9) | 1.46 (1.00–2.13) | 0.048 | 292 (47.5) | 1.22 (0.81–1.84) | 0.351 | 394 (64.1) | 1.55 (1.08–2.23) | 0.017 |
| Functional Illiteracy | No | 170 (24.1) | 1 | | 262 (37.2) | 1 | | 382 (54.2) | 1 | |
| | Yes | 75 (33.2) | 0.72 (0.47–1.09) | 0.122 | 154 (68.1) | 1.25 (0.83–1.88) | 0.296 | 158 (69.9) | 1.04 (0.67–1.60) | 0.878 |
| Ethnicity | Others | 120 (26.7) | 1 | | 224 (49.8) | 1 | | 261 (58.0) | 1 | |
| | Mestizo | 125 (26.0) | 1.04 (0.75–1.45) | 0.798 | 192 (39.9) | 0.78 (0.55–1.10) | 0.158 | 279 (58.0) | 1.21 (0.90–1.64) | 0.213 |
| Occupation | Agricultural workers | 55 (21.1) | 1 | | 111 (42.5) | 1 | | 142 (54.4) | 1 | |
| | Household chores | 139 (31.9) | 1.78 (0.87–3.66) | 0.116 | 227 (52.1) | 3.67 (1.71–7.90) | 0.001 | 288 (66.1) | 1.54 (0.80–2.95) | 0.194 |
| | Non-agricultural workers | 51 (21.8) | 1.74 (1.01–2.98) | 0.045 | 78 (33.3) | 1.92 (1.13–3.26) | 0.017 | 110 (47.0) | 1.24 (0.78–1.98) | 0.370 |
| Recent alcohol consumption | No | 122 (35.0) | 1 | | 198 (56.7) | 1 | | 234 (67.0) | 1 | |
| | Yes | 120 (20.9) | 0.76 (0.52–1.10) | 0.146 | 216 (37.6) | 0.80 (0.57–1.12) | 0.191 | 301 (52.3) | 0.91 (0.66–1.26) | 0.573 |
| Current smoker | No | 229 (26.7) | 1 | | 388 (45.2) | 1 | | 503 (58.6) | 1 | |
| | Yes | 16 (22.2) | 1.08 (0.49–2.38) | 0.851 | 28 (38.9) | 0.98 (0.77–1.26) | 0.9 | 37 (51.4) | 0.83 (0.44–1.56) | 0.561 |
| Physically demanding work | No | 162 (26.8) | 1 | | 269 (44.5) | 1 | | 350 (57.9) | 1 | |
| | Yes | 81 (25.4) | 1.13 (0.77–1.67) | 0.53 | 142 (44.5) | 1.10 (0.75–1.61) | 0.637 | 185 (58.0) | 0.88 (0.61–1.26) | 0.476 |
| Contact with agrochemicals | No | 185 (27.3) | 1 | | 310 (45.7) | 1 | | 397 (58.6) | 1 | |
| | Yes | 60 (23.7) | 0.93 (0.61–1.42) | 0.741 | 106 (41.9) | 0.72 (0.48–1.10) | 0.126 | 143 (56.5) | 0.90 (0.60–1.36) | 0.631 |
| **History of chronic diseases or risk factors** | | | | | | | | | | |
| Vascular events | No | 217 (25.6) | 1 | | 360 (42.5) | 1 | | 485 (57.3) | 1 | |
| | Yes | 27 (33.3) | 1.22 (0.71–2.08) | 0.467 | 55 (67.9) | 2.45 (1.38–4.35) | 0.002 | 54 (66.7) | 1.35 (0.74–2.46) | 0.330 |
| Hypertension | No | 110 (17.7) | 1 | | ------ | ------ | ------ | ------ | ------ | ------ |
| | Yes | 134 (43.9) | 2.08 (1.44–3.01) | 0.001 | ------ | ------ | ------ | ------ | ------ | ------ |
| Diabetes | No | ------ | ------ | ------ | 312 (39.2) | 1 | | ------ | ------ | ------ |
| | Yes | ------ | ------ | ------ | 103 (77.4) | 3.01 (1.79–5.07) | <0.001 | ------ | ------ | ------ |
| Elevated cholesterol | No | 114 (20.2) | 1 | | 213 (37.8) | 1 | | 284 (50.4) | 1 | |
| | Yes | 130 (35.7) | 1.71 (1.23–2.36) | 0.001 | 203 (55.5) | 1.56 (1.08–2.24) | 0.018 | 255 (70.1) | 1.62 (1.15–2.27) | 0.005 |
| **Chronic diseases (history and/or clinical measurements)** | | | | | | | | | | |
| Hypertension | No | 79 (15.3) | 1 | | ------ | ------ | ------ | ------ | ------ | ------ |
| | Yes | 166 (39.9) | 2.09 (1.40–3.11) | <0.001 | ------ | ------ | ------ | ------ | ------ | ------ |
| Diabetes | No | ------ | ------ | ------ | 250 (36.4) | 1 | | ------ | ------ | ------ |
| | Yes | ------ | ------ | ------ | 166 (67.8) | 1.95 (1.31–2.90) | 0.001 | ------ | ------ | ------ |
| **Nutritional and laboratory measures** | | | | | | | | | | |
| Obesity | Not overweight (BMI<25) | 42 (13.0) | 1 | | 93 (28.9) | 1 | | 80 (24.8) | 1 | |
| | Overweight (BMI 25–29) | 113 (30.8) | 2.29 (1.48–3.56) | <0.001 | 179 (48.8) | 1.99 (1.28–3.10) | 0.002 | 248 (67.6) | 5.85 (3.86–8.88) | <0.001 |
| | Obese (BMI≥30) | 90 (37.2) | 3.25 (1.96–5.40) | <0.001 | 144 (59.5) | 4.20 (2.55–6.93) | <0.001 | 212 (87.6) | 27.6 (15.8–48.2) | <0.001 |

*(Continued)*

Table 3. (Continued)

| Risk factor | Category | Type-2 Diabetes | | | Hypertension | | | Metabolic Syndrome | | |
|---|---|---|---|---|---|---|---|---|---|---|
| | | n (%) | OR (95% CI) | P value | n (%) | OR (95% CI) | P value | n (%) | OR (95% CI) | P value |
| Abdominal obesity | No | 44 (11.6) | 1 | | 94 (24.7) | 1 | | ------ | ------ | ------ |
| | Yes (men≥94 cm; women≥88 cm) | 201 (36.5) | **2.85 (1.85–4.39)** | **<0.001** | 322 (58.4) | **3.12 (2.15–4.54)** | **<0.001** | ------ | ------ | ------ |
| Elevated cholesterol | No | 103 (21.2) | 1 | | 174 (35.8) | 1 | | 241 (49.6) | 1 | |
| | Yes (≥200 mg/dL) | 142 (31.9) | 1.09 (0.77–1.54) | 0.629 | 252 (54.4) | 1.28 (0.89–1.84) | 0.175 | 299 (67.2) | 1.37 (0.98–1.92) | 0.069 |
| Elevated triglycerides | No | 99 (19.9) | 1 | | 186 (37.4) | 1 | | ------ | ------ | ------ |
| | Yes (≥150 mg/dL) | 146 (33.6) | **1.66 (1.20–2.32)** | **0.003** | 230 (53.0) | 1.26 (0.90–1.76) | 0.172 | ------ | ------ | ------ |
| Low HDL | No | 93 (22.9) | 1 | | 174 (42.8) | 1 | | ------ | ------ | ------ |
| | Yes (men<40 mg/dL; women<50 mg/dL) | 152 (29.0) | 1.32 (0.93–1.87) | 0.126 | 242 (42.8) | 1.40 (0.97–2.02) | 0.069 | ------ | ------ | ------ |
| Insulin resistance | No | 46 (11.9) | 1 | | 123 (31.9) | 1 | | 127 (33.0) | 1 | |
| | Yes (HOMA>2.5) | 199 (36.4) | **3.77 (2.41–5.90)** | **<0.001** | 293 (53.7) | **3.05 (2.10–4.42)** | **<0.001** | 413 (75.6) | **8.77 (6.15–12.51)** | **<0.001** |

Odd ratios (ORs) and 95% confidence intervals were estimated using logistic regression and accounting for household and community clustering and weighted for population structure. Age$^2$ was strongly associated with all 3 outcomes: type-2 Diabetes (OR 0.9986, 95% CI 0.9982–0.9990, P = 0.001), hypertension (OR 0.9993, 95% CI 0.9988–0.9999, P = 0.028), and metabolic syndrome (OR 0.9992, 95% CI 0.9986–0.9997). P<0.001). Associations between outcomes and risk exposures, clinical history, and laboratory measures were adjusted for the sociodemographic variables shown in Table 1 (including age and age$^2$ [all outcomes], sex [all outcomes], and the interaction between age and sex [hypertension and the metabolic syndrome]). Functional illiteracy was defined as 3 or fewer years of formal schooling [31]. Outcome definitions: i) Type 2 diabetes—clinical history of diabetes treatment and/or glycosylated hemoglobin (HbA1c) ≥6.5%; ii) hypertension–clinical history of antihypertensive treatment and/or ≥140 mm Hg or diastolic blood pressure ≥90 mm Hg during 2 separate measurements [21]; and iii) metabolic syndrome–using the Harmonized criteria [22]; and iv) vascular events- self-reported history of heart attack or stroke. Abdominal obesity was measured using waist circumference. Missing data: alcohol consumption (7); current smoker (1); physical demanding work (8); history of: vascular events (3), respiratory disease (3), kidney disease (3), elevated cholesterol (3), diabetes (3) and hypertension (3). Missing data: alcohol consumption (n = 7); current smoker (1); physical demanding work (8); history of vascular events (3); diabetes (3) and hypertension (3); and bioimpedance (increased fat %) (205). Abbreviation: BMI–body mass index; HDL–high-density lipoprotein; HOMA—Homeostatic Model Assessment index.

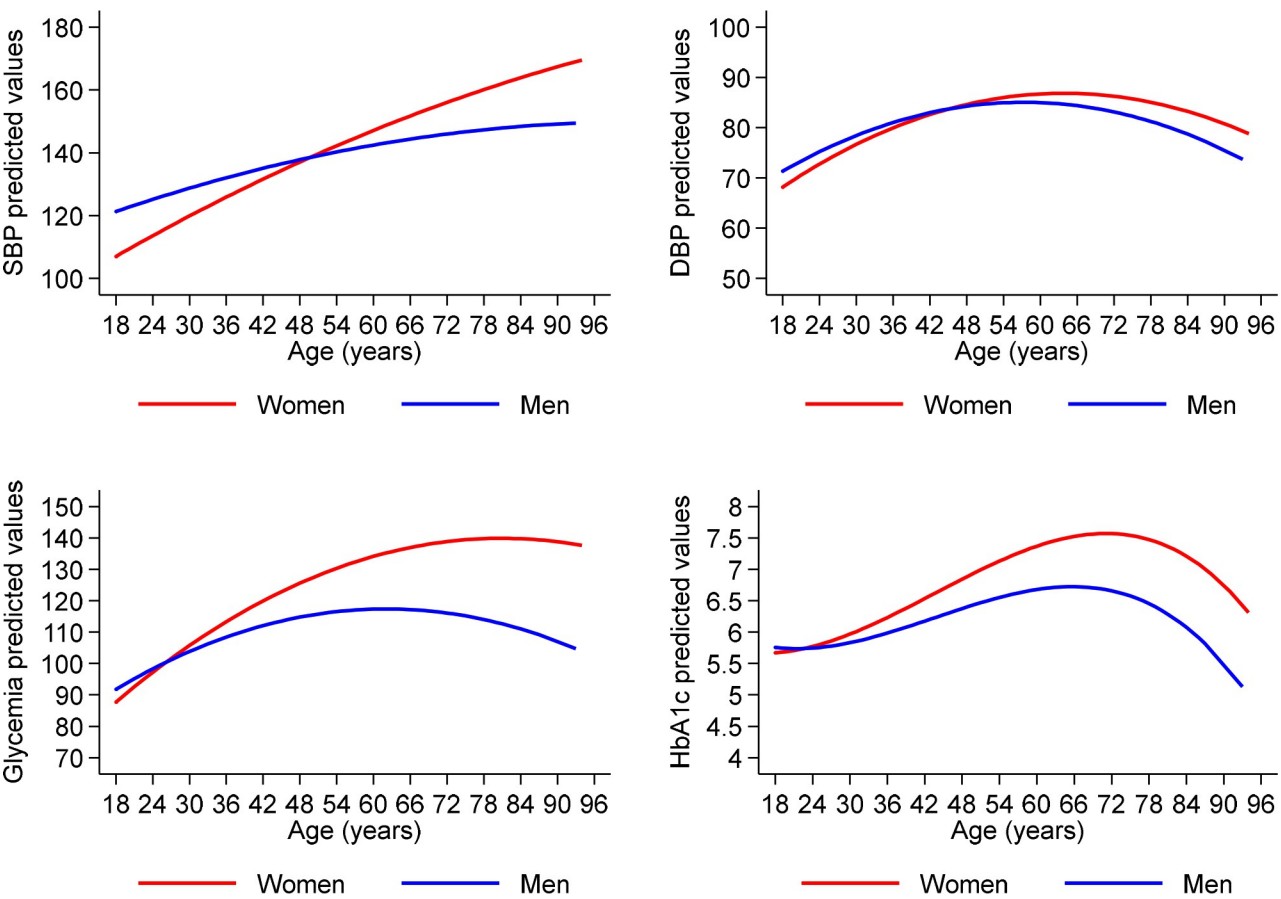

**Fig 3. SBP, DBP, glycemia and HbA1c predicted levels stratified by gender.** Predictions were from regression on age, gender and their potential interactions accounting for family cluster and weighted for population characteristics.

Weighted prevalence of T2D, HTN and MetS observed here was greater than might be expected from the findings of previous surveys from Ecuador [29, 32, 33] and elsewhere in Latin America [34–36]. Previous studies have estimated T2D prevalence of 8.7% in the Latin American region [34] and 4.7% in adult Ecuadorians [37]. The most recent national survey from 2018 estimated a T2D prevalence of 7.1%, indicating a likely temporal trend of increasing prevalence among Ecuadorian adults [32]. Other studies have shown that marginalized population groups, particularly indigenous populations, appear to be particularly vulnerable to high rates of T2D: 70% of studies in indigenous populations evaluated in a recent systematic review reported prevalence greater than 10% [38]. A recent study of Afro-Ecuadorian adults living in conditions of severe poverty in a rural region of coastal Ecuador, estimated a T2D prevalence of 6.8% [39]. Although differences in estimated prevalence between studies could be explained partly by differences in disease definitions, mean age of study population, and study design, our estimate of 20.4% was unexpectedly high.

The age-standardized prevalence of HTN declined over the period 1975–2015 in HICs while tending to increase in many LMICs with an overall doubling of numbers of people with HTN globally because of population growth and ageing [6]. Prevalence of HTN in Latin America and the Caribbean declined over the same period [6] although age-standardized prevalence remained high (35.4%) in the region in 2019 [35]. Our estimate of 35.6% HTN prevalence is close to this value and is similar to that reported among Afro-Ecuadorians (36%), another

marginalized group living in rural transitional coastal communities [15] among whom HTN was considered the most important cause of death [40]. However, our estimate of HTN prevalence was more than double that reported among adults living in urban and rural settings elsewhere in the country [29, 41, 42].

MetS is known to predict sudden death [43] and increases markedly the risk of cardiovascular diseases [44]—even one to two components of the syndrome have been associated with increased mortality [45, 46]. A population-based survey of adults from 7 Latin American cities including Quito estimated prevalence of MetS ranging from 14% in Quito to 27% in Barquisimeto, Venezuela [36]. Within Latin America a high prevalence of MetS has been reported among indigenous populations including 46.7% in the Amazon region of Venezuela [47] and 37% in Brazil, although in Brazil the prevalence in 6 surveys of Indigenous groups varied markedly between 11% in Parana and 66% in Matto Grosso [48]. Our estimate of 54.2% was similar to that obtained (55.7%) in a house-to-house survey of adults living in a (Cholo-)Montubio community in the southern coastal region of Ecuador [49], but greater than that reported (42%) in urban and rural areas in the central Andean region of the country [50] and in a national survey (31.2%) [33]. Differences in prevalence estimates within Ecuador could represent differences in definitions, sampling methods, as well as differing population risks for individual components of the syndrome.

Rural communities undergoing rapid changes relating to urbanization processes are of particular relevance for studies examining potential causal links between changes in risk behaviors, development of risk factors, and emergence of cardiometabolic disease. The coastal region of Ecuador, where we did this study, is of particular interest because of unique geographic, cultural, and socio-economic characteristics, likely to affect distributions of risk factors and NCD prevalence. Montubios are recognized in Ecuador as a distinct mestizo-derived ethnicity and represent a historically excluded group living in rural, often low mountainous and isolated regions (<500 m) of coastal Ecuador [51, 52]. Montubio communities adhere closely to traditional values, being a patriarchal culture based on tight kinship ties, conservative values, and a close relationship with the land. However, Montubio communities are presently undergoing a rapid transition to a more westernized lifestyle [53], and many communities are found in ecologically vulnerable environments. Traditionally, the inhabitants of these communities lived as agriculturalists (as day-laborers or cultivators of smallholdings) with seasonal work on large coffee plantations. The region consists of mountainous terrain currently undergoing rapid degradation relating to deforestation and soil erosion, resulting in poorly fertile soils for agriculture. There is a high vulnerability to rapid runoff and landslides and the area has been designated as of extreme ecologic risk by the local municipal authorities, and excluded from national and local government investments in infrastructure and basic services including health care. These factors likely have contributed to a process of acculturation, likely accelerated by occupational shifts from the land to services generally provided outside the communities; the effects of the migration of the young in search of employment either temporarily or permanently; and with the introduction of electricity, the impact of technological innovations such as television and digital media. Such changes have occurred over a period of less than a generation (road access and electricity were introduced 15–20 years previously) and have been accompanied by increased sedentarism and a shift in diet to include increasing quantities of processed foods high in fat, sugar, and salt [29, 54] that can be acquired cheaply in local stores.

Several studies have shown rural populations in LMICs to have healthier cardiometabolic profiles than urban residents, an effect attributed to greater physical activity more than dietary effects [55–57]. The sedentarization of rural occupations likely has led to changes in cardiometabolic risk factors profiles and disease risk to resemble urban populations. The prevalence of

vascular events (8.7% with stroke or heart attack) observed here is somewhat lower than reported from another Montubio population further south in Coastal Ecuador where stroke prevalence increased from 14.1% to 35.2% between 2003 and 2012 [49, 58]. Possible explanations are poorer survival rates or the more recent emergence of an unhealthy profile of cardiometabolic risk factors in our population. Environmental degradation may have compounded such effects through more rapid shifts in occupation, diets, and activity levels. Rapid changes in diet and physical activity affecting a population exposed to chronic under-nutrition in childhood likely will saturate metabolic capacity in adulthood [59] and increase vulnerability to the premature development of cardiometabolic risk factors and disease.

Indigenous groups in Latin America appear to be at particular risk of cardiometabolic NCDs especially diabetes [38] in the context of acculturation, deforestation, and environmental degradation [17]. Other rural marginalized groups might be expected to suffer similar risks although available data are limited [60]. An additional challenge for these populations is inadequate health care access which further contributes to the burden of disability and premature death. Lack of access to health care results in delays in diagnosis and limits treatment access. Even where there is health access, a lack of education and financial resources often limits availability of and adherence to treatment which often needs to be provided regularly for life. This was seen in the present study by the observation that greater than 65% of those with a previous doctor diagnosis of T2D or HTN had poorly controlled glucose or blood pressure, respectively. Lower socioeconomic conditions in LMIC populations have been consistently linked to cardiometabolic NCD prevalence as well as to MetS or its individual components [3, 39, 61–63]. A rational public health strategy for NCD control and prevention in resource-poor rural populations will require community-based interventions, ideally through primary health care (or community health worker networks) where available and targeted to the specific risk factors and pathologies prevalent in specific populations [64, 65].

## Limitations

Data collection targeted a marginalized population living in a restricted geographic setting where we believed there to be major unmet health needs. The data can be generalized to similar populations living in coastal Ecuador in conditions of extreme ecologic and social vulnerability. Non-participation of healthier members of the communities could have led to overestimation of estimates, likely related to more active, healthier individuals being away for work. Survey non-participation was largely explained by absence of working-age adults (20–40 years), particularly men, because of salaried work commitments outside the communities. Under-representation of adults aged 20–40 years that was apparent based on a comparison of age-structure between our house-to-house-census and the projected census population for the district, was adjusted for using weighted estimates. The use of questionnaires to collect data on clinical histories and risk factors may be subject to recall or social desirability bias although we used standardized clinical and laboratory measurements to measure risk factors and outcomes where possible.

## Conclusion

We observed a high prevalence of T2D, HTN, and MetS in marginalized Montubio communities in a rural region of coastal Ecuador. Environmental degradation and accelerated urbanization occurring in these transitional communities over a period of approximately 20 years, with increased out-migration, occupational shifts, and changes in diet and physical activity are likely to be important causes of these NCDs. Our data indicate that marginalized rural populations undergoing a rapid transition from traditional to more modern lifestyles in Ecuador and

elsewhere in Latin America, suffer an unacceptable burden of preventable morbidity. This burden might be reduced by targeted public health strategies including the training and adequate resourcing of community health workers.

## Supporting information

**S1 Fig. Age distribution in the study sample compared to that of the Portoviejo population, stratified by gender.** Data were obtained from the 2010 census [9].
(TIF)

**S1 Table. Clinical history of diabetes and presence of hyperglycemia (Hb1Ac> = 6.5%) at time of survey.**
(DOCX)

**S2 Table. Clinical history of hypertension and presence of elevated blood pressure (>140/90 mm Hg) at time of survey.**
(DOCX)

**S3 Table. Age-and sex adjusted associations between indicators of adiposity and systolic and diastolic blood pressure (mm Hg), glycosylated hemoglobin (HbA1c) (%), and fasting glucose (mg/dL) in 931 adults.** Estimates and 95% confidence intervals (CI) were derived from linear regression analyses and accounted for household and community clustering and were weighted for population structure. Associations, i.e. the adjusted mean differences between groups defined by indicators of adiposity were adjusted for the sociodemographic variables shown in the Table (including age and age$^2$ [all outcomes], age$^3$ [HbA1c and fasting glucose], sex [all outcomes], and the interaction between age and sex [all outcomes]). Abbreviations: BP–blood pressure; WHtR–waist-to-height ratio; VAI–Visceral adiposity index. Definitions for indicators of adiposity: Overweight (BMI≥25); abdominal obesity (waist circumference of ≥94 cm in men and ≥88 cm in women); increased body fat (men (≥25%, women (≥30%); WHtR (≥0.5); elevated VAI (age <30 –VAI >2.52; age ≥30 & <42 – VAI > 2.23; age ≥42 & <52 –VAI >1.92; age ≥52 & <66 –VAI > 1.93; age ≥66 –VAI > 2). Missing data: increased fat (205).
(DOCX)

**S1 Data. Raw data used for analyses.**
(TXT)

## Acknowledgments

The authors thank participant communities and their representatives for their co-operation, and the support of technicians and health professionals from Universidad Internacional del Ecuador and the CAMERA study.

## Author Contributions

**Conceptualization:** Peter Whincup, Natalia Romero-Sandoval, Philip J. Cooper.

**Data curation:** Monsermin Gualan, Irina Chis Ster.

**Formal analysis:** Irina Chis Ster.

**Funding acquisition:** Natalia Romero-Sandoval, Philip J. Cooper.

**Investigation:** Monsermin Gualan, Tatiana Veloz, Emily Granadillo, Luz M. Llangari-Arizo, Alejandro Rodriguez, Julia A. Critchley, Miguel Martin, Natalia Romero-Sandoval, Philip J. Cooper.

**Methodology:** Peter Whincup, Natalia Romero-Sandoval, Philip J. Cooper.

**Project administration:** Emily Granadillo, Natalia Romero-Sandoval, Philip J. Cooper.

**Resources:** Natalia Romero-Sandoval, Philip J. Cooper.

**Supervision:** Natalia Romero-Sandoval, Philip J. Cooper.

**Visualization:** Irina Chis Ster, Alejandro Rodriguez.

**Writing – original draft:** Monsermin Gualan, Philip J. Cooper.

**Writing – review & editing:** Irina Chis Ster, Tatiana Veloz, Emily Granadillo, Luz M. Llangari-Arizo, Alejandro Rodriguez, Julia A. Critchley, Peter Whincup, Miguel Martin, Natalia Romero-Sandoval.

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
