## [Decision Letter · Decision Letter 0]

9 Apr 2024

PONE-D-24-03952Prevalence and risk factors for type-2 diabetes, hypertension, and metabolic syndrome among adults living in marginalized rural communities in tropical coastal EcuadorPLOS ONE

Dear Dr. Cooper,

Thank you for submitting your manuscript to PLOS ONE. After careful consideration, we feel that it has merit but does not fully meet PLOS ONE’s publication criteria as it currently stands. Therefore, we invite you to submit a revised version of the manuscript that addresses the points raised during the review process.

We look forward to receiving your revised manuscript.

Kind regards,

Neftali Eduardo Antonio-Villa, MD PhD

Academic Editor

PLOS ONE

Journal Requirements:

"Universidad Internacional del Ecuador (grant EDM-INV-04-19)"

**Additional Editor Comments:**

Monsermin Gualan et al. conducted a cross-sectional study to evaluate the prevalence of diabetes, hypertension, and metabolic syndrome (MS) in a marginalized rural zone in the tropical coastal area of Ecuador. The authors sampled 931 individuals using diverse questionnaires and collected anthropometrical and biochemical samples. They described a high prevalence of diabetes, hypertension, and MS (affecting over half of the sample) and evaluated associated factors. They found an association between adiposity metrics, non-agricultural work, and these health outcomes. This study offers significant public health insights for Ecuador, but several aspects require clarification.

Abstract

• Recommend changing “higher values” to “higher prevalence” for clarity.

• Suggest removing the mention of previous CVD prevalence, as it is unrelated to the study's objectives.

Introduction

• The introduction is well-written. I would suggest that strengthening the justification for this study would enhance its impact. Specifically, elucidate how the study's findings can benefit indigenous and vulnerable populations whose characteristics have been diminishing.

Methods

• Clarify whether fasting glucose levels >126 mg/dl were considered in the T2D diagnosis.

• Justify the choice of >190 mg/dl as the threshold for elevated cholesterol, differing from the AHA guideline of ≥200 mg/dl.

• Recommend splitting the “definitions” paragraph into two: one for “outcome definitions” (T2D, arterial hypertension, and MS) and another for defining other conditions, such as adiposity indicators.

• When estimating the age and sex-adjusted effects of variables on outcomes, clarify if “effects” refers to adjusted prevalences.

Results

• For Figure 2, include the 95% confidence intervals (CI) for each outcome, possibly adjusting for age and sex via log binomial regression to more accurately model outcome probabilities. Reference: https://www.bookdown.org/rwnahhas/RMPH/blr-log-binomial.html

• In the “Population-weighted prevalence of diabetes, hypertension, metabolic…“paragraph, the authors first describe the prevalence of T2D, then HTN, and lastly, MS, but in Figure 2, these are displayed in reverse order. I would suggest arranging them into a sequence described in the results section.

• Clarify whether the associations in Table 3 derive from a saturated model or represent bivariate adjustments.

• Consider relocating Table 5 to the Supplementary Material.

Discussion

• An excellent discussion.

Figures and Tables

• In Figure 1, it would be optimal to include a scale for each transition in territory.

• In Figure 2, please change the scale of the y-axis to be interpreted on a 100% scale instead of a proportion.

Reviewers' comments:

Reviewer's Responses to Questions

**Comments to the Author**

1. Is the manuscript technically sound, and do the data support the conclusions?

Reviewer #1: Yes

Reviewer #2: Yes

Reviewer #3: Yes

2. Has the statistical analysis been performed appropriately and rigorously? 

Reviewer #1: Yes

Reviewer #2: Yes

Reviewer #3: Yes

3. Have the authors made all data underlying the findings in their manuscript fully available?

Reviewer #1: Yes

Reviewer #2: Yes

Reviewer #3: No

4. Is the manuscript presented in an intelligible fashion and written in standard English?

Reviewer #1: Yes

Reviewer #2: Yes

Reviewer #3: Yes

5. Review Comments to the Author

Reviewer #1: Dear Authors,

-My main question is "not clear the novelty of the study". Considering numerous published studies from Ecuador, and NCD data of 2020 (Ref.11), data of this sub-national study is interesting for local Journals, and their results cannot generalize to the country population.

-What is your mention of this sentence "evaluation of participants between 6 and 10 am"?

-Please clarify method/measurement of visceral adiposity index, body fat, HOMA-index, and LDL in the method.

-For prediction, did you just adjust with age and sex? Where are ROC cure, sensitivity and specificity?

Minor revisions:

-Nearly 30% of the Ref. are published in more than 10 years ago. Please update.

-Tables S1, and S2 can be removed.

Best Regards,

Reviewer #2: Gualan et al performed a cross-sectional study looking at the prevalence of cardiometabolic disease and risk factors in an Ecuadorian rural population. Authors concluded there was an unexpectedly high prevalence of diabetes, hypertension and metabolic syndrome.

The manuscript is mostly well written. There’s certainly a gap in the care of rural communities and the need to understand more what their risk factors are.

The novelty of this manuscript is the population that was studied.

Comments and suggestions:

1. Line 147 “sex, sex and age” needs to be fixed

2. As part of the questionnaire, was “regular exercise” asked about?

3. Smoking is a known risk factor for the development of hypertension and diabetes, can you include reasons why your think there wasn’t an association in this cohort?

4. 20.4% prevalence of T2DM is very high, even compared to other studies done in rural communities in Ecuador. I will recommend elucidating more in the discussion why the authors think that is the case. Maybe the criteria used to define diabetes by using “clinical history of diabetes treatment” was a bit inaccurate as it may also be including people with prediabetes. We know some patients when they have prediabetes may be started on metformin for example. It will be relevant to include the percentage of patients with an “objective” diagnosis of diabetes based on HbA1c of 6.5% or greater.

5. There’s a study looking at the prevalence of diabetes in other rural areas in Ecuador published in July of 2023

Puig-García M, Caicedo-Montaño C, Márquez-Figueroa M, Chilet-Rosell E, Montalvo-Villacis G, Benazizi-Dahbi I, Peralta A, Torres-Castillo AL, Parker LA. Prevalence and gender disparities of type 2 diabetes mellitus and obesity in Esmeraldas, Ecuador: a population-based survey in a hard-to-reach setting. Int J Equity Health. 2023 Jul 1;22(1):124. doi: 10.1186/s12939-023-01939-x

It will be worth mentioning it in the discussion – compare/contrast. This study also found higher prevalence of diabetes in women but the overall prevalence of diabetes was lower than the one described in this study.

6. HbA1c should also be shown as categorial variable, A1c >= 6.5% and <6.5%

Reviewer #3: I have had the opportunity to review the manuscript titled " Prevalence and risk factors for type-2 diabetes, hypertension, and metabolic syndrome among adults living in marginalized rural communities in tropical coastal Ecuador ", which presents a cross-sectional study aimed at reporting the prevalence of cardiovascular diseases such as diabetes, metabolic syndrome, and hypertension in rural communities of a coastal province in Ecuador, characterised by the socio-cultural traits of the Montubio population in the area. The researchers unveil concerning prevalence rates for all these conditions, particularly for metabolic syndrome, and surprisingly high rates of diabetes compared to national surveys and other territories.

I believe this work addresses a fundamental issue from a quite insightful perspective, taking into consideration the obesogenic context in which individuals live, subjected to structural inequalities in access to resources and healthcare, subsequently impacting health inequities. The authors do a great work exposing these determinants in the discussion of the paper. Therefore, in my opinion, this study would be suitable for publication in PLOS ONE, as it provides essential information for the development of public policies addressing these issues in hard-to-reach communities in low- and middle-income countries such as Ecuador.

However, I think it is important to make a series of minor changes before its publication, which I have listed below.

Line 90 � Communities were selected by convenience sampling. Censuses for each community were drawn up by community representatives: How was this contact to the community done? Was there a previous relationship between the authors and the population? Were the censuses updated for the study? Consider providing further information regarding the sample, the community representatives and the extend of their role in the study.

Line 100 � Data collection: Following the previous point, who took part in the collection of data? Did it involve local health promotors or just the study team?

Line 120 � Definitions: a) Consider including the categorisation of the risk factors such as occupation and ethnicity (do “not mestizos” include white and indigenous in the same category?). It would facilitate the reading of the manuscript, because it is only mentioned afterwards in the results. b) Line 136-137: cardiovascular disease is shown in tables disaggregated as diabetes, hypertension and hypercholesterolemia, maybe you should indicate it in this section to clarify the understanding of the tables (you show e.g. hypertension as the measurement + history, and then only as history.

Lines 147-148 � “stratified by sex, sex and age (…) and their interaction where (when would be more correct) statistically significant”.

Line 161 � Of 1,525 adults eligible according to the house-to-house census, 1,010 adults were recruited from 10 communities, and 931 (61.0% of those eligible) had complete(d) data (collection) and were included in the analysis.

Line 167 � alcohol and tobacco have important differences in consumption between men and women, consider to desegregate.

Line 169 � were any of those adiposity indicators different enough between men and women to consider mention a couple of them in the text?

Lines 189-191� Prevalence of HTN increased with age, although there was some evidence of an interaction with sex, with a particularly steep increase in prevalence in women (consider to reformulate to increase clearness: particularly showing a steep increase…)

Lines 193-196 � Some commas are missing: Prevalence of MetS was similar in both sexes up to about 40 years, after which prevalence was greater in women. An analysis of 576 participants, for whom blood pressure could be repeated at home, showed a lower average systolic blood pressure in the home setting.

Line 211� I am not sure why you only mention some of the risk exposures that show a significant p-value, for example for DM2: hypertension (history, consider using HTN in the table next to the hypertension in chronic diseases), lives with a partner, obesity, triglycerides and insulin resistance are also significant. If you only mention some relevant, please indicate in the text.

Line 258 � Table 5 (keep using the same p value, sometime .001 /0.001 // age (-0.004 (-0.007 - 0.0??)) // try to keep the minus next to the number -0.451 (-0.589 - -

0.313)

Line 393 � Adjusted for/by?? for seems to be a mistake

Lines 400-403 � “20 years or so” does not look like a good fit for an academic text. consider the following idea to improve the academic writing: “Environmental degradation and accelerated urbanization occurring in these transitional communities over a period of approximately 20 years, with increased out-migration, occupational shifts, and changes in diet and physical activity are likely to be important causes of cardiovascular diseases.”

Lines 403-407 � Rephrase the last paragraph as it is not completely clear and suggests that the data show a preventable morbidity that could be effectively targeted. // more modern ways of living (lyfestyles?) // mitigated through (by) targeted public health strategies

While the discussion is well organised and rich, helping to understand the problem of this hard-to-reach areas with great detail and sense of understanding, the conclusion needs some improvement, as mentioned before. I appreciate the opportunity to review this manuscript, congrats for your work.

6. PLOS authors have the option to publish the peer review history of their article (what does this mean?). If published, this will include your full peer review and any attached files.

Reviewer #1: No

Reviewer #2: No

Reviewer #3: No

---

## [Author Response · Author response to Decision Letter 0]

28 May 2024

'Response to Editors and Reviewers'.

Comment 1. Please ensure that your manuscript meets PLOS ONE's style requirements, including those for file naming. The PLOS ONE style templates can be found at 

Response 1: We have ensured that the manuscript now meets PLOS ONE’s style requirements including those for file naming.

Comment 2: Thank you for stating the following financial disclosure: 

"Universidad Internacional del Ecuador (grant EDM-INV-04-19)"

Response 2: We confirm that the statement is correct and have included the statement in the cover letter.

Comment 3. Please note that in order to use the direct billing option the corresponding author must be affiliated with the chosen institute. Please either amend your manuscript to change the affiliation or corresponding author, or email us at plosone@plos.org with a request to remove this option.

Response 3: We have updated the affiliation of the corresponding author’s primary institution, St George’s University of London. 

Comment 4. When completing the data availability statement of the submission form, you indicated that you will make your data available on acceptance. We strongly recommend all authors decide on a data sharing plan before acceptance, as the process can be lengthy and hold up publication timelines. Please note that, though access restrictions are acceptable now, your entire data will need to be made freely accessible if your manuscript is accepted for publication. This policy applies to all data except where public deposition would breach compliance with the protocol approved by your research ethics board. If you are unable to adhere to our open data policy, please kindly revise your statement to explain your reasoning and we will seek the editor's input on an exemption. Please be assured that, once you have provided your new statement, the assessment of your exemption will not hold up the peer review process.

Response 4: We have provided the data used for the analysis as a supplementary ‘.txt’ file and provided a data availability statement at the end of the manuscript.

Comment 5. Please include your full ethics statement in the ‘Methods’ section of your manuscript file. In your statement, please include the full name of the IRB or ethics committee who approved or waived your study, as well as whether or not you obtained informed written or verbal consent. If consent was waived for your study, please include this information in your statement as well.

Response 5: We have amended the ethics statement as requested in the ‘Methods’ section as follows: “Informed written consent was obtained from each participant and the study protocol was approved by Bioethics Committee of the UTE University (Comité de Ética de Investigación en Seres Humanos de la Universidad UTE, CEISH UTE 2019-1121-03) and by the Ecuadorian Ministry of Public Health (MSP-DIS-2021-0393-O)”.

Comment 6: Additional Editor Comments: Monsermin Gualan et al. conducted a cross-sectional study to evaluate the prevalence of diabetes, hypertension, and metabolic syndrome (MS) in a marginalized rural zone in the tropical coastal area of Ecuador. The authors sampled 931 individuals using diverse questionnaires and collected anthropometrical and biochemical samples. They described a high prevalence of diabetes, hypertension, and MS (affecting over half of the sample) and evaluated associated factors. They found an association between adiposity metrics, non-agricultural work, and these health outcomes. This study offers significant public health insights for Ecuador, but several aspects require clarification.

Response 6: We thank the editor for their positive review of the manuscript

Comment 7. Abstract

• Recommend changing “higher values” to “higher prevalence” for clarity.

• Suggest removing the mention of previous CVD prevalence, as it is unrelated to the study's objectives.

Response 7: Corrected as requested.

Comment 8. Introduction

• The introduction is well-written. I would suggest that strengthening the justification for this study would enhance its impact. Specifically, elucidate how the study's findings can benefit indigenous and vulnerable populations whose characteristics have been diminishing.

Response 8: We have added the following sentence to the end of the introduction: “Our findings, showing a high prevalence of cardiometabolic diseases, highlight an unmet need for community-based public health strategies to minimize the growing burden of premature death and morbidity from NCDs among marginalized and indigenous populations living in transitional rural settings.”

Comment 9. Methods

• Clarify whether fasting glucose levels >126 mg/dl were considered in the T2D diagnosis.

Response 9: Fasting glucose was not used to define T2D diagnosis. The T2D definition for this study was a clinical history of diabetes treatment and/or glycosylated hemoglobin (HbA1c) ≥6.5% (page 7, lines 135-136).

Comment 10. • Justify the choice of >190 mg/dl as the threshold for elevated cholesterol, differing from the AHA guideline of ≥200 mg/dl.

Response 10: Thank you for observing this error. The data have now been corrected for elevated total cholesterol using a cut-off of ≥200 mg/dl.

Comment 11. • Recommend splitting the “definitions” paragraph into two: one for “outcome definitions” (T2D, arterial hypertension, and MS) and another for defining other conditions, such as adiposity indicators.

Response 11: Corrected as requested.

Comments 12. • When estimating the age and sex-adjusted effects of variables on outcomes, clarify if “effects” refers to adjusted prevalences.

Response 12: We have rephrased this sentence of the statistical analysis section for clarity as follows: “We estimated age and sex adjusted effects of explanatory variables of interest on prevalence of these outcomes”

Comment 13. Results

• For Figure 2, include the 95% confidence intervals (CI) for each outcome, possibly adjusting for age and sex via log binomial regression to more accurately model outcome probabilities. Reference: https://www.bookdown.org/rwnahhas/RMPH/blr-log-binomial.html

Response 13: We have now added the 95% confidence intervals of the predictions as requested. The 95% CIs are fairly robust, with the exception of the tail of the age which reflects relatively sparse observations for old people. As for log binomial regression, we are not entirely sure why it would produce more accurate probabilities. The model applies to aggregated type data, namely when the data is limited to the number of cases and the total number in age groups for example. We hold individual level data and that is the most appropriate method to apply to such data. The model takes into account age and sex and accommodates for weights derived from population data such that the curves are generalizable. Moreover, we have accounted for the variability induced by communities and households the participants belong to. A log binomial model applied to groups of 1 can easily run in convergence problems – we do not think that there is any theoretical reason for which this would produce more accurate results. 

Comment 14. • In the “Population-weighted prevalence of diabetes, hypertension, metabolic…“paragraph, the authors first describe the prevalence of T2D, then HTN, and lastly, MS, but in Figure 2, these are displayed in reverse order. I would suggest arranging them into a sequence described in the results section.

Response 14: We changed the order of graphs in Fig 2 to T2D, HTN, and MS, as requested. 

Comment 15: • Clarify whether the associations in Table 3 derive from a saturated model or represent bivariate adjustments.

Response 15: We are not entirely sure about the rationale of this question. In the context of a logistic regression a saturated model is a perfect prediction model, namely a model which predicts 1/0 only. The associations in Table 3 are adjusted effects (of a series of relevant variables) for sex and age (together with up its 3rd power terms, namely age, age2 and age3).

Comment 16: • Consider relocating Table 5 to the Supplementary Material.

Response 16: Table 5 has now been moved to supplementary material as S3 Table

Comment 17: Discussion

• An excellent discussion.

Figures and Tables

• In Figure 1, it would be optimal to include a scale for each transition in territory.

Response 17: A km scale has now been added to each transition in territory.

Comment 18: • In Figure 2, please change the scale of the y-axis to be interpreted on a 100% scale instead of a proportion.

Response 18: The y-axis of Figure 2 has now been changed to a 100% scale.

Comment 19: Reviewer #1: Dear Authors,

-My main question is "not clear the novelty of the study". Considering numerous published studies from Ecuador, and NCD data of 2020 (Ref.11), data of this sub-national study is interesting for local Journals, and their results cannot generalize to the country population.

Response 19: The novelty of this study, as highlighted by another reviewer (please see comment 24) lies in the study of a marginalized rural population of Montubios in coastal Ecuador. This population differs from the dominant mestizo population in terms of sociocultural and other characteristics, discrimination, and living in poor rural environments where they are particularly vulnerable to the health consequences of environmental degradation and recent and rapid urbanization. There are limited published epidemiological studies of cardiometabolic diseases of marginalized rural populations in Latin America, often because of difficulties and costs of data collection and community participation. We tried to address this gap in the literature using a strategy based on community participation in which community representatives were involved the decision-making, planning and implementation processes of the study. This approach of dialogue and community participation would likely have led to a better understanding of these prevalent conditions and the unmet need for improved access to health care, early diagnosis, and timely treatment to prevent long-term complications. Having said that, we understand that the editorial policies of PLOS ONE are to place a greater emphasis on scientific rigour and ethics than perceived novelty.

Comment 20: -What is your mention of this sentence "evaluation of participants between 6 and 10 am"?

Response 20: We have specified timing to show that data and sample collection was done at approximately the same time of day for all participants given that diurnal periodicity of some measurements could lead to potential bias, and earlier sampling was essential for working in agricultural communities and to ensure maintenance of fasting. 

Comment 21: -Please clarify method/measurement of visceral adiposity index, body fat, HOMA-index, and LDL in the method.

Response 21: Method for measurement of visceral adiposity index, body fat, and HOMA-index have now been clarified in the Methods. Visceral adiposity index was calculated using a model of adipose distribution corrected for triglyceride and HDL cholesterol levels (Reference 23); body fat was estimated using bioimpedance based on the equation provided in Reference 27: in the absence of a population-validated equation, this equation was used to estimate fat-free mass; HOMA index was calculated using fasting glucose and insulinemia as described in References 28 and 29 This information has been added to the Methods (lines 153-175). HDLc was measured using enzymatic colorimetry assays (lines 139-140)– we did not estimate LDL.

Comment 21: For prediction, did you just adjust with age and sex? Where are ROC cure, sensitivity and specificity?

Response 21: The short answer is yes, the predictions are across age (together with up its 3rd power terms, namely age, age2 and age3) and sex only. The purpose of this study was exactly this, to understand patterns in the population in this health outcomes according to age and sex, and to understand the adjusted effects of other factors (Table 3). With respect to ROC curves: We did not aim at constructing a predictive most parsimonious model and investigate its performance as a classification model. This was an epidemiological study with clearly set a priori objectives and it would be wrong for us to speculation on such a model when not planned for. This is a relatively restricted population, and the study is in essence exploratory. The analysis accommodates for the hierarchical structure of the data and the population (age and sex)-derived weights makes the resulting patterns of the probabilities of these conditions by age and sex generalizable to the population from which they were derived. 

Comment 22: Minor revisions:

-Nearly 30% of the Ref. are published in more than 10 years ago. Please update.

Response 22: We have updated references as requested. Most of the older references are original references for specific methodologies, data sources, or theories to which more recent references refer. 

Comment 23: Tables S1, and S2 can be removed.

Response 23: We believe that S1 and T2 Tables in the supplementary information provide valuable information on undiagnosed and inadequately treated disease in this population with limited access to health care and would prefer to keep these tables in the supplement information.

Comment 24. Reviewer #2: Gualan et al performed a cross-sectional study looking at the prevalence of cardiometabolic disease and risk factors in an Ecuadorian rural population. Authors concluded there was an unexpectedly high prevalence of diabetes, hypertension and metabolic syndrome.

The manuscript is mostly well written. There’s certainly a gap in the care of rural communities and the need to understand more what their risk factors are.

The novelty of this manuscript is the population that was studied.

Response 24: We thank the reviewer for highlighting the novelty of this study and the knowledge gap of studies done in rural communities.

Comment 25: Line 147 “sex, sex and age” needs to be fixed.

Response 25: Corrected as requested.

Comment 26: As part of the questionnaire, was “regular exercise” asked about?

Response 26: We did not ask about regular exercise but rather the questions were focused on physically demanding work which we believe would be more appropriate for inhabitants of rural and agricultural communities. 

Comment 27: Smoking is a known risk factor for the development of hypertension and diabetes, can you include reasons why your think there wasn’t an association in this cohort?

Response 27: Current smoking was relatively infrequent (7.7%) in the study population so there may have been an issue of study power although ORs for outcomes were close to 1. Smoking as a risk factor appears to be infrequently reported in studies of cardiometabolic diseases in adults from Ecuador possibly because of the low frequency of this exposure (eg Reference 55) – 1 study that did evaluate smoking in Ecuadorian adults did not report associations of metabolic syndrome or diabetes with smoking (Reference 47).

Comment 28: 20.4% prevalence of T2DM is very high, even compared to other studies done in rural communities in Ecuador. I will recommend elucidating more in the discussion why the authors think that is the case. Maybe the criteria used to define diabetes by using “clinical history of diabetes treatment” was a bit inaccurate as it may also be including people with prediabetes. We know some patients when they have prediabetes may be started on metformin for example. It will be relevant to include the percentage of patients with an “objective

---

## [Decision Letter · Decision Letter 1]

16 Jun 2024

PONE-D-24-03952R1Prevalence and risk factors for type-2 diabetes, hypertension, and metabolic syndrome among adults living in marginalized rural communities in tropical coastal EcuadorPLOS ONE

Dear Dr. Cooper,

Thank you for submitting your manuscript to PLOS ONE. The manuscript could be accepted for publication after addressing minor comments made by one revisor. Therefore, we invite you to submit a revised version of the manuscript that addresses the points raised during the review process.

We look forward to receiving your revised manuscript.

Kind regards,

Neftali Eduardo Antonio-Villa, MD PhD

Academic Editor

PLOS ONE

Journal Requirements:

Additional Editor Comments:

Dear authors,

Thank you for addressing the comments made by all reviewers.

One reviewer made additional valuable comments that must be addressed before making a final decision.

Reviewers' comments:

Reviewer's Responses to Questions

**Comments to the Author**

1. If the authors have adequately addressed your comments raised in a previous round of review and you feel that this manuscript is now acceptable for publication, you may indicate that here to bypass the “Comments to the Author” section, enter your conflict of interest statement in the “Confidential to Editor” section, and submit your "Accept" recommendation.

Reviewer #1: (No Response)

Reviewer #2: All comments have been addressed

Reviewer #3: All comments have been addressed

2. Is the manuscript technically sound, and do the data support the conclusions?

Reviewer #1: Yes

Reviewer #2: Yes

Reviewer #3: Yes

3. Has the statistical analysis been performed appropriately and rigorously? 

Reviewer #1: Yes

Reviewer #2: Yes

Reviewer #3: Yes

4. Have the authors made all data underlying the findings in their manuscript fully available?

Reviewer #1: Yes

Reviewer #2: Yes

Reviewer #3: Yes

5. Is the manuscript presented in an intelligible fashion and written in standard English?

Reviewer #1: (No Response)

Reviewer #2: Yes

Reviewer #3: Yes

6. Review Comments to the Author

Reviewer #1: Dear Authors,

The quality of manuscript is significantly improved.

Please consider following minor revisions:

-Consider language editing in whole manuscript

-Brief title and change it to an interesting tile such as:

Cardiometabolic diseases and their associated risk factors in transitional rural communities of Ecuador

-Expand words in the first use, for example STEPs

-Correct 'sera" in line 121.

-Insert unit of all variables in the tables.

-Merge table 4 and 1.

-Fig S1, not necessary. Plz omit.

-Recheck number of table S3, it is table 5 or S3?

Best Regards,

Reviewer #2: (No Response)

Reviewer #3: All comments have been adequately addressed following suggestions and corrected minor revisions. The data analyzed in the study are provided in the supporting information. Congratulations on your work.

7. PLOS authors have the option to publish the peer review history of their article (what does this mean?). If published, this will include your full peer review and any attached files.

Reviewer #1: No

Reviewer #2: No

Reviewer #3: No

---

## [Author Response · Author response to Decision Letter 1]

1 Jul 2024

Comment 1: Reviewer #1: Dear Authors,

The quality of manuscript is significantly improved.

Please consider following minor revisions:

-Consider language editing in whole manuscript

Response 1: The manuscript has now been reviewed and edited by two native English speakers.

Comment 2: -Brief title and change it to an interesting tile such as:

Cardiometabolic diseases and their associated risk factors in transitional rural communities of Ecuador

Response 2: We thank the reviewer for this suggestion. We have now changed the title to: “Cardiometabolic diseases and associated risk factors in transitional rural communities in tropical coastal Ecuador”

Comment 3: -Expand words in the first use, for example STEPs

Response 3: We have now expanded acronyms/abbreviations on first definition and deleted others where not essential. 

Comment 4: -Correct 'sera" in line 121.

Response 4: ‘Sera’ has been changed to ‘serum’.

Comment 5: -Insert unit of all variables in the tables.

Response 5: Units for variables have been added to Tables.

Comment 6: -Merge table 4 and 1.

Response 6: We are wondering if the reviewer is referring to Table 2 rather than Table 1? Table 1 shows raw data in which no inferences have been made with respect to the hierarchical nature of the data. Table 4 (and Table 2) provides inferential estimates accounting for the hierarchical structure of the data and the age structure of the sample with respect to the census population. We have now combined Tables 2 and 4 as a single table (new Table 2). 

Comment 7: -Fig S1, not necessary. Plz omit.

Response 7: While we agree that this supplementary Figure may not be necessary for many readers, we do believe that it would be informative for some interested readers as it instructs on the derivations of the weights obtained using the census data. The effects of weighting can be seen in the weighted estimates for study outcomes and in the 95% CIs for means of blood pressure, fasting glucose, and HbA1c. The figure thus provides useful information to allow the interested reader to understand how the census data was used (and can be used for future studies) after data collection to derive representative estimates

Comment 8: -Recheck number of table S3, it is table 5 or S3?

Response 8: Title of S3 table has been corrected.

---

## [Editor Report · Decision Letter 2]

4 Jul 2024

Cardiometabolic diseases and associated risk factors in transitional rural communities in tropical coastal Ecuador

PONE-D-24-03952R2

Dear Dr. Cooper,

We’re pleased to inform you that your manuscript has been judged scientifically suitable for publication and will be formally accepted for publication once it meets all outstanding technical requirements.

Kind regards,

Neftali Eduardo Antonio-Villa, MD PhD

Academic Editor

PLOS ONE

Additional Editor Comments (optional):

Dear authors,

All minor comments were appropriately addressed, and an extensive English edition was produced. I can now process and recommend the acceptance of the manuscript in its current form. It has been a true honor to review and manage this manuscript. It will contribute to understanding prevalent cardiometabolic risk factors in Ecuador.
---

## [Editor Report · Acceptance letter]

10 Jul 2024

PONE-D-24-03952R2 

PLOS ONE

Dear Dr. Cooper, 

I'm pleased to inform you that your manuscript has been deemed suitable for publication in PLOS ONE. Congratulations! Your manuscript is now being handed over to our production team.

Kind regards, 

on behalf of

Dr. Neftali Eduardo Antonio-Villa 

Academic Editor

PLOS ONE